# Interactions between Guidance Cues and Neuronal Activity: Therapeutic Insights from Mouse Models

**DOI:** 10.3390/ijms24086966

**Published:** 2023-04-09

**Authors:** Maitri Tomar, Jamie Beros, Bruno Meloni, Jennifer Rodger

**Affiliations:** 1School of Biological Sciences, The University of Western Australia, Crawley, WA 6009, Australia; 2Perron Institute for Neurological and Translational Science, Nedlands, WA 6009, Australia; 3Centre for Neuromuscular and Neurological Disorders, The University of Western Australia, Crawley, WA 6009, Australia; 4Department of Neurosurgery, Sir Charles Gairdner Hospital, QEII Medical Centre, Nedlands, WA 6009, Australia

**Keywords:** topography, ephrin, sensory systems, neuronal activity, rTMS, neuroplasticity

## Abstract

Topographic mapping of neural circuits is fundamental in shaping the structural and functional organization of brain regions. This developmentally important process is crucial not only for the representation of different sensory inputs but also for their integration. Disruption of topographic organization has been associated with several neurodevelopmental disorders. The aim of this review is to highlight the mechanisms involved in creating and refining such well-defined maps in the brain with a focus on the Eph and ephrin families of axon guidance cues. We first describe the transgenic models where ephrin-A expression has been manipulated to understand the role of these guidance cues in defining topography in various sensory systems. We further describe the behavioral consequences of lacking ephrin-A guidance cues in these animal models. These studies have given us unexpected insight into how neuronal activity is equally important in refining neural circuits in different brain regions. We conclude the review by discussing studies that have used treatments such as repetitive transcranial magnetic stimulation (rTMS) to manipulate activity in the brain to compensate for the lack of guidance cues in ephrin-knockout animal models. We describe how rTMS could have therapeutic relevance in neurodevelopmental disorders with disrupted brain organization.

## 1. Introduction

The mammalian brain is made up of billions of neurons that connect to each other to form an elaborate and sophisticated network. Neurons establish precise point-to-point connections between multiple regions in the brain, forming a topographic map to maintain their spatial order [1]. This map provides structural and functional organization to several brain regions [1] and is an important step in development as it is crucial for representing and integrating different sensory inputs [2,3]. Topographic maps are established and refined during the developmental period, and abnormal topography has been linked to disorders such as autism spectrum disorder (ASD) and schizophrenia, where there are deficits in sensory processing [4,5]. Therefore, the establishment of correct brain organization is important for the healthy functioning of the brain, but how is such a complex formation achieved? This is a result of multiple complex processes, such as axon guidance cues and neuronal activity working together to build and refine neural circuits during development [3].

Developing neurons sense and navigate their environment using growth cones, specialized structures at the tip of axons, and dendrites, which detect and respond to molecular guidance cues to reach their target location in the brain [6]. The neuronal activity also plays a key role in growth cone guidance and is necessary to refine projections in response to sensory and other functional inputs [7]. There are four major families of axon guidance molecules in the brain: netrins, semaphorins, slits, and ephrins and their receptors [8]. These molecular guidance cues can either be attractive or repulsive. In the presence of attractive cues, growth cones stabilize, whereas, in the presence of repulsive cues, they retract or collapse, which steers developing axons to their target location in the brain [8]. Some guidance cues also play a role in the organization of the projection of an entire population of neurons by providing spatial information through their expression patterns. The best-known example of this is the Eph and ephrin families of axon guidance molecules, which are the focus of this review.

The role of Eph-ephrin signaling in forming maps in the brain was predicted by a study in which it was suggested that mapping of retinal axons to the optic tectum could be most parsimoniously achieved by two sets of orthogonal molecular gradients [9]. This hypothesis led to the search for molecular gradients in the retinal and visual brain centers and resulted in the identification of ephrin ligands and their Eph receptors [10]. Ephrins steer developing axons to their target via attractive and repulsive guidance molecules and are involved in short-range contact-mediated cell-cell interaction and bidirectional signaling [11]. Ephrins can be divided into two classes based on their mode of attachment to the cell membrane; ephrin-A ligands are attached via a glycosyl-phosphatidylinositol (GPI) linkage, while ephrin-B ligands are transmembrane proteins [11]. Ephrins bind selectively to Eph receptors from the same class, with the exception of the EphA4 receptor that binds to ephrin-As as well as both ephrin-B2 and B3 [11]. Since these initial findings, several studies have manipulated the expression and distribution of ephrin-A and their receptors in animal models and transgenic mouse lines to better understand the role of these guidance cues in the representation of sensory information in the brain. EphB and ephrin-B families also have important roles in visual system development (particularly axonal decussation [12] and dorso-ventral patterning [13]) and in the hippocampus [14,15,16,17,18]. However, in this review, we focus on the ephrin-A family because of their key role in topographic mapping and integration of topographic projections.

In this review, we will describe selected transgenic models and how they have been utilized to study both structural and functional roles of EphA-ephrinA signaling in different sensory systems. We will also discuss the behavioral consequences of abnormal formation of topographic maps in the brain when Eph-ephrin signaling is disrupted. Mice lacking Eph-ephrin guidance cues have offered unexpected insights into the role of neuronal activity in axon guidance, particularly by allowing experiments that manipulate neuronal activity to rescue abnormal organization and function across multiple brain circuits. We conclude by describing experiments that use transgenic animal models to test approaches to treating neurological disorders involving abnormal connectivity.

## 2. Ephrins in Sensory Systems

### 2.1. Visual System

The visual pathway begins in the eye, formed by retinal ganglion cells (RGCs) located in the retina and their efferent projections to various brain targets (Figure 1). The formation of this circuit begins early in development where in mice, contralateral retinal projections reach the superior colliculus (SC) in the midbrain and dorsal geniculate nucleus of the thalamus (dLGN) around embryonic day 16 (E16), while the smaller number of ipsilateral projections arrive later in embryonic development between E18 and birth [19]. Ipsilateral projections from the dLGN then project to the primary visual cortex (V1), which then project to the ipsilateral SC for the integration of retinal and cortical inputs [20,21]. There are many other target regions within the visual pathway [22], but these are beyond the scope of this review and are not included in the present brief overview.

During normal development in mice, RGCs enter the contralateral rostral SC at E16 and initially overshoot their targeted termination zone (TZ). The overshoot is then retracted, and dense TZ arborizations are formed at the appropriate location [23]. Although this pattern is well described in the literature, it is interesting to note that RGC birthdate during the embryonic period influences the accuracy of their target selection [24], and this may also extend to their response to other factors that modulate axon guidance processes [25,26].

Retraction of the overshooting growth cones is dependent on ephrin guidance cues [27]. One of the earliest studies investigating this in the chick visual system described the graded expression patterns of ephrin-A2 and -A5 in the tectum [10]. The expression of ephrin-A2 and -A5 increased from anterior to posterior (rostral to caudal) in the tectum, and this corresponded with decreasing temporal to nasal EphA3 expression in the retina [10]. In this way, temporal retinal axons with high expression of EphA receptors are repulsed from the caudal SC because of the high level of ephrin-A ligands, while nasal retinal axons, which have low levels of EphA receptors, are less sensitive to ephrin-A ligands and therefore terminate in the caudal SC (Figure 1). Similar mapping processes have been observed in other regions of the visual system, such as the dLGN, where ephrin-A2 and -A5 ligands also have a graded expression pattern, while ephrin-A3 expression is low [28] (Figure 1). While the graded expression pattern of Eph and ephrins throughout the visual system is strongly suggestive of their key role in the spatial organization of visual projections, direct evidence has come from studies of a range of knockout (KO) and knock-in (KI) transgenic animal models.

#### 2.1.1. Knockout Animal Models

##### Collicular Pathways

In transgenic animal models of single or multiple ephrin gene deletions, there are disruptions along the retinocollicular and corticocollicular pathways. Using anatomical tracing, it has been shown that single mutants, ephrin-A2^−/−^ and ephrin-A5^−/−^ mice, have minor defects along the retinocollicullar map, identified as single ectopic TZs of temporal and nasal retinal axons in the contralateral caudal and rostral SC, respectively [29,30]. In ephrin-A5^−/−^ mice, ectopic arborizations were found in rostral and caudal SC but not in mid-SC, where ephrin-A2 expression is high [29]. Similar abnormalities are found in the ipsilateral retinal projection, but this will be discussed later in the review in the context of the integration of projections. Also, in the absence of ephrin-A5 specifically, contralateral retinal axons overshoot into the inferior colliculus (IC; generally, a region of high ephrin-A5 expression), although these projections are not maintained in adults [29]. The ectopic projections likely represent the persistence of overshooting growth cones or aberrant branches that occur during normal development and are normally removed by Eph-ephrin signaling [27]. However, it has not been confirmed whether the ectopic terminations originate from individual RGCs or result from aberrant branching [31]. Unsurprisingly, mapping defects are more severe in double (ephrin-A2^−/−^/A5^−/−^ mice) and triple mutants (ephrin-A2^−/−^/A3^−/−^/A5^−/−^ mice), where a greater number of diffuse ectopic TZs are found in the corticocollicular, retinogeniculate and retinocollicular projections [30,32,33]. Moreover, these ectopic arborizations are also found in the medio-lateral axis, which is not as severe as the disruptions found along the rostro-caudal axis of the SC [30,32,33]. This was suggested to be a secondary effect of having disruptions along the rostro-caudal axis of SC as during development, the sensory map along the medio-lateral axis in the SC (dorsoventral axis of the visual field) is established after reading the projections along the rostro-caudal axis (nasal-temporal axis of the visual field) [23,30]. Therefore, ephrin-As are important in establishing topography along both axes in the SC.

##### Thalamic Pathways

Ephrin-KO models have also been used to study changes along the retinogeniculate and thalamocortical pathways [32,34,35,36]. Compared to the SC, in which 2-dimensional retinal projections are mapped across a 2-dimensional surface, the organization within the dLGN is more complex to quantify, as terminations are mapped across a 3-dimensional structure [33]. In addition, the presence of both retinotopic organization and eye-specific laminae increases the challenge of quantitative analyses of spatial location. Nonetheless, the principles of graded expression in both regions are similar: within the dLGN, ephrin-A2 and A5 are expressed in a high to low expression pattern in the ventral-lateral-anterior to dorsal-medial-posterior gradient, whereas ephrin-A3 has a lower expression in dLGN [28]. In mice lacking ephrin-As, abnormal mapping is observed across the anterior-posterior axis of dLGN, where nasal retinal projections extend farther anteriorly compared to the projections from the temporal retina [33]. However, these disruptions in the retinogeniculate projections are less severe compared to retinocollicular projections and do not include ectopic TZs [32,33].

Mapping has also been studied in the visual cortex of mice lacking ephrin-As. Unlike the dLGN and SC, which are both subcortical structures located deep within the brain, the visual cortex in rodents is superficial and located directly beneath the skull. This characteristic permits the use of intrinsic signal imaging in live animals to map the functional organization of the visual input [37]. Using this approach, severe disruptions in the thalamocortical projections were identified along the medial-lateral axis of V1 (nasal-temporal retinal axis) in triple ephrin-KO mice (ephrin-A2/A3/A5^−/−^) [34]. The visuotopic representation was compressed and rotated along the nasal-temporal retinal axis and medially shifted within the V1. This shift along the medial-lateral axis and not the anterior-posterior axis in the visual cortex was suggested to be due to a lack of repulsion from adjacent regions of the V1 on the dLGN axonal projections [34]. Interestingly, the medial shift of the visual cortex was only significant for ephrin-A2/A3/A5^−/−^ mice when compared to wild-type mice and was not observed in double-KOs (ephrin-A2/A5^−/−^, A3/A5^−/−^ or A2/A3^−/−^). This phenotype is similar to the overshooting of retinal axons into the IC in the absence of ephrin-A5 [29] and further confirms that the repulsion mediated by ephrin-As is involved in defining boundaries between brain regions. It also suggests specific roles for individual ephrin-As in the distinct features of organization, orientation, and positioning of the input to V1.

##### The Role of Neuronal Activity

In addition to the genetic modification of guidance cues described above, the activity of RGCs has also been manipulated in order to investigate the role of activity-dependent mechanisms in mapping. In mice, retinal activity is present from very early in development (E16), even before eye-opening, in the form of waves of spontaneous RGC activity mediated by cholinergic and glutamatergic neurotransmission [38]. These waves of activity propagate across the retina, reinforcing correct neighbor relationships between RGCs and their terminals within the SC: these activity-dependent processes refine the spatial information carried by the gradients of Eph receptors and ephrin ligands [39,40]. Around the time of birth, cholinergic retinal waves are involved in refining retinotopic maps throughout the visual system, segregating eye-specific inputs to the dLGN [28], and may also play a role in aligning corticocollicular and retinocollicular maps [41]. After birth, the waves switch from cholinergic to glutamatergic and continue to assist with retinotopic refinement and eye-specific segregation in the last stages of spontaneous retinal activity before visual input begins [42].

The importance of these retinal waves for topographic map formation was demonstrated in mice in which both ephrin-As and the β2 subunit of the nicotinic acetylcholine receptor (nAChR) was ablated. Triple-KO ephrin-A2/A3/A5^−/−^ mice, which have essentially no ephrin signaling in their visual system, but intact retinal waves, show severe errors in the refinement of axonal arborization which are much more serious than double- or single-KOs, although these mice still retain overall topography [33]. It was only when the genes encoding both ephrin-As and β2 nAChR were deleted (triple-KO mice lacking ephrin-A2/A5 and β2 nAChR) that retinotopy in the SC and visual cortex were essentially abolished, with extensive topographic targeting errors and ectopic terminations persisting throughout the developmental period [33,43]. These studies are crucial because they show that although guidance cues and patterned neuronal activity can individually provide enough spatial information to establish some topography, both systems are required for normal topographic maps.

The ability to manipulate the contributions of activity and ephrins separately in these models also allows the mechanisms of eye-specific mapping and segregation in the dLGN to be probed. Using anatomical tracing, severe defects were found in the topography of both contralateral and ipsilateral retinal projections to the dLGN of both ephrin-A2/A5^−/−^ and ephrin-A2/A3/A5^−/−^ mice [28]. Ipsilaterally projecting RGCs extended axons along the entire dorsoventral axis of the dLGN, whereas in wild-type mice, the projections were limited to the dorsomedial quadrant. This defect was more severe in ephrin-A2/A3/A5^−/−^ mice suggesting that ephrin-A3, in particular, is involved in the guidance of ipsilateral projections to their appropriate location in the dLGN. Unlike in the retinocollicular pathway [44], the inappropriately located ipsilateral patches in the dLGN were not due to the greater number of ipsilateral projections into the dLGN of ephrin-A2/A3/A5^−/−^ mice [28]. Despite the abnormal topography, the segregation of the two inputs remained intact. However, when cholinergic retinal activity was blocked by epibatidine, there was significant overlap between TZs of ipsilateral and contralateral projections, mostly in the dorsal dLGN, confirming the key role of activity and not ephrin-As in eye-specific segregation [28]. These experiments of multiple ephrin-A gene KOs have definitively demonstrated that topography and eye-specific projections involve different mechanisms and suggest that ephrin-As and activity have independent roles in mapping the dLGN.

However, there is evidence that there are direct signaling interactions between ephrin and activity-dependent mechanisms. Activity-mediated retraction of overshooting retinal axons in response to ephrin-A5 is dependent on secondary messengers such as calcium and cyclic adenosine monophosphate (cAMP) [27,45]. Response of growth cones to ephrin-A5 is dependent on the presence of calcium-stimulated adenylate cyclase 1 (AC1) in the retina, and this interaction could be dependent on cAMP signaling [27]. Using retinal explants, a study showed that blocking spontaneous activity in retinal explants using tetrodotoxin reduced the collapse and retraction speed of growth cones in response to ephrin-A5, which was rescued by periodic oscillations in cAMP [45]. It was suggested that following neuronal depolarization, AC1 might be responsible for causing transient changes in cAMP and subsequent calcium release to initiate growth cone collapse in response to ephrin-A [45]. Therefore, ephrin ligands, together with spontaneous activity and nonsynaptic mechanisms, are crucial for the targeting and refinement of retinal projections and eye-specific segregation in the thalamus.

#### 2.1.2. Knock-In Animal Models

In an elegant approach to understanding the role of graded Eph-ephrin signaling in visual system topography, Brown et al. [46] mapped retinocollicular projections in an EphA3 knock-in (EphA3-KI) mouse model at postnatal day 11–12 [43]. The approach was highly original and significant because although previous transgenic approaches have removed Eph/ephrin proteins, the global impact of the manipulation was not able to address specifically the role of the gradient that had been predicted previously [9]. Overexpression of EphA receptors only in RGCs expressing the Islet2 gene resulted in two spatially overlapping populations of RGCs, each expressing a discrete gradient [46]. These two intermingled populations mapped to two continuous but distinct maps in the SC: EphA receptor overexpressing RGC axons projected more rostrally than neighboring RGCs that did not overexpress EphA receptors. In addition, temporal retinal axons that did not overexpress EphA receptors terminated in mid-SC instead of rostral SC, suggesting that a retinal axon will change its projection to a more caudal location if it encounters another RGC that expresses a higher level of EphA receptor. Therefore, relative EphA expression and axon-axon competition are important for map formation in the visual system.

In conclusion, the visual system has historically been the focus of studies of topographic organization, probably because of early work [9] that anticipated the role of graded Eph and ephrin expression. However, topography is a commonly observed principle in the brain, and it is therefore not surprising that the same guidance cues are used across other brain regions that require the organization of inputs from various sensory areas.

### 2.2. Somatosensory Cortex

The primary somatosensory cortex (S1) in mice, and particularly the whisker barrel fields, have been a very popular model system to study connectivity, particularly because the whiskers are arguably more ethologically relevant to rodents. The whiskers send sensory information from the face to the brainstem, then to the thalamus, and finally to layer IV of S1 or the barrel cortex [47] (Figure 2). Analogous to the representation of the visual field within the visual pathway, the organization of whiskers into rows and columns is maintained in the organization of their projections throughout the brainstem and thalamus to their cortical target, the whisker barrel cortex [47]. It is, therefore, not surprising that graded expression of Ephs and ephrins has been implicated in mapping thalamic projections to S1 in mice [35,48,49].

Ephrin-A5 is expressed in a medial-to-lateral gradient across S1, whereas EphA4 receptors are expressed in the ventrobasal complex of the thalamus in a complementary gradient [49] (Figure 2). Direct evidence for the involvement of Eph/ephrin signaling came from a study of ephrin-A5^−/−^ mice, which have altered barrel dimensions in the posteromedial barrel subfield (PMBSF) of S1 [49]. Although the overall thalamocortical map and the number of PMBSF barrels were normal, there was a contraction in medial barrels, but the expansion of lateral barrels was evident during development and maintained in the adults. The decrease in inter-barrel distance and individual barrel area could be due to the lack of repellent signaling in medial PMBSF (the area of maximal ephrin-A5 expression in wild-type mice), resulting in a medial shift of the thalamocortical map. The functional representations of these distorted barrel fields were further investigated using intrinsic signal optical imaging to measure whisker-evoked activity in S1 of ephrin-A5^−/−^ mice [48]. It was found that even though individual whisker representation and point-to-point mapping between the thalamus and S1 were not disrupted, the whisker functional representations were compressed and overlapping.

The phenotype of the somatosensory map in ephrin-A5^−/−^ mice differs from that of the visual system in that the somatosensory map retains overall organization, while the visual maps are much more disrupted. In addition, the overlap in the functional representation of the whiskers in the somatosensory map is reminiscent of visual maps that develop with abnormal activity-dependent processes, suggesting that ephrins may interact more with activity in the somatosensory map. In agreement with this, ephrin-As were found to be involved in the elimination of motor cortical dendritic spines, which is an important developmental process for the stabilization and maturation of neural circuits in the brain and one that is regulated by neuronal activity [50,51]. Accelerated spine elimination in the motor cortex was seen in ephrin-A2^−/−^ mice, but following unilateral whisker trimming, there was a significant reduction in spine elimination in the barrel cortex of both wild-type and ephrin-A2^−/−^ mice with no effect on synapse formation suggesting that ephrin-A2 is involved in experience-dependent pruning of the synapses by spine loss but does not regulate spine formation [52]. Interestingly, the accelerated spine elimination observed in the barrel cortex of ephrin-A2^−/−^ mice was also dependent on the activation of N-methyl-D-aspartate (NMDA) receptors, similar to the action of sensory experience [52].

After the visual system, the somatosensory cortex is another sensory system that has been extensively studied to identify the role of Eph-ephrin signaling in topographic mapping. However, unlike the visual system, the topographic errors along the thalamocortical projections in the somatosensory cortex in the absence of ephrin ligands are minor [48,49]. But the studies on the role of ephrin-A in the somatosensory cortex have revealed how these ligands interact with activity to regulate synaptic pruning, which suggests its important role in the formation of mature neural circuits during development.

### 2.3. Auditory System

Ephrin expression is well-defined in the visual system, and the somatosensory cortex, but the expression and function of this ligand family in the auditory system are not well characterized. Ephrin-A2 and -A5 have been found in various regions of the developing and mature auditory system in rodents, such as the auditory nerve, cochlea including the cochlear nerve cells, cochlea connective tissue, supporting inner and outer hair cells, and in deeper and more central structures such as the IC and medial geniculate nucleus [53,54,55,56,57,58]. Ephrin-A3 is highly expressed in the lateral areas of the cochlea and the vestibular system [54,56].

Ephrins play an important role in the auditory brainstem as deletion of ephrin-A2 and -A5 results in an increased sensitivity to sound, which is more prominent in double mutant ephrin-A2/A5^−/−^ mice [59]. These mice show a lower threshold for high-frequency stimuli compared to wild-type mice which means that for a given frequency, neurons in the auditory brainstem of ephrin-A2^−/−^ and ephrin-A2/A5^−/−^ mice generate a stronger response [59]. On the other hand, ephrin-A5^−/−^ mice had normal thresholds. However, they show a reduction in wave I amplitude; a decrease in peripheral auditory nerve function due to reduced input from inner hair cells [59]. This is similar to the condition of tinnitus in humans, where, despite reduced wave I amplitude and abnormal hearing, there is still normal auditory brainstem function [60].

This change in cochlear function as a result of ephrin-A5 deletion can be explained by a study [55] that reported expression patterns and projection abnormalities that are homologous to those reported in the visual system of ephrin-KO mice, suggesting similar functional roles. During normal (wild-type) development, EphA4 expression was detected in type I spiral ganglion neurons (SGNs), while ephrin-A5 was strongly expressed in the outer hair cells [55] (Figure 3A). Furthermore, receptor-ligand binding resulted in repulsion, demonstrated in vitro by ephrin-A5-induced growth cone collapse of type I SGNs through activation of the EphA4 receptor. The resulting complementary gradients and functional interactions of the Eph receptor and ephrin ligand are similar to the patterns described in the visual system [10,28]. In ephrin-A5^−/−^ mice, type I SGNs of the cochlea that typically innervate the inner hair cells overshoot (due to the lack of inhibitory ephrin-A5) and reach the outer hair cells [55]. On the other hand, type II SGNs that extensively innervate the outer hair cells send out much shorter projections to this region in ephrin-A5^−/−^ mice, presumably due to increased competition from the overshooting type I neurons [55].

Errors in decussation were also seen during the development of the auditory system in ephrin-A2/A5^−/−^ mice [61]. In wild-type mice, axonal projections from the ventral cochlear nucleus (VCN) target the contralateral medial nucleus of the trapezoid body (MNTB) in the brainstem. However, in ephrin-A2/A5^−/−^ mice, although contralateral projections are normal, there are numerous aberrant ipsilateral projections [61]. This pattern was seen not only in double but also in single ephrin-A2- or A5-KO mice. Interestingly, despite errors in VCN projection laterality, all three strains have normal VCN-MNTB topography where the projections from dorsoventral VCN ended in mediolateral MNTB.

In contrast to ephrin-A2 and -A5 ligands which are involved in laterality/decussation and potentially boundary formation, ephrin-A3 is more specifically involved in mapping auditory system topography [62]. This is facilitated by complementary graded expression of receptors EphA4 and A7 in the SGNs and of the ephrin-A3 ligand in the cochlear nucleus (CN), which guides SGNs to the developing cochlea [62]. Early in development, the future high-frequency auditory nerve fibers innervate dorsal CN (DCN) as they are repelled by the high expression of ephrin-A3 in the VCN. However, future low-frequency auditory nerve fibers, which have low sensitivity towards ephrin-A3, invade the VCN. In the absence of ephrin-A3, there is no repulsion to the initial high-frequency auditory nerve fibers, and these fibers terminate more ventrally than expected, which then hinders the growth of low-frequency auditory nerve fibers into the VCN. This results in an abnormal tonotopic map in ephrin-A3^−/−^ mice, and these mice show impairment in sound discrimination.

The graded expression of Eph receptors and their ephrin ligands to promote topography in the auditory system is similar to what we have discussed above for the visual and somatosensory systems. However, the auditory system has shown the distinct roles of different ephrin-A ligands in defining boundaries, laterality, and topography.

**Figure 3 ijms-24-06966-f003:**
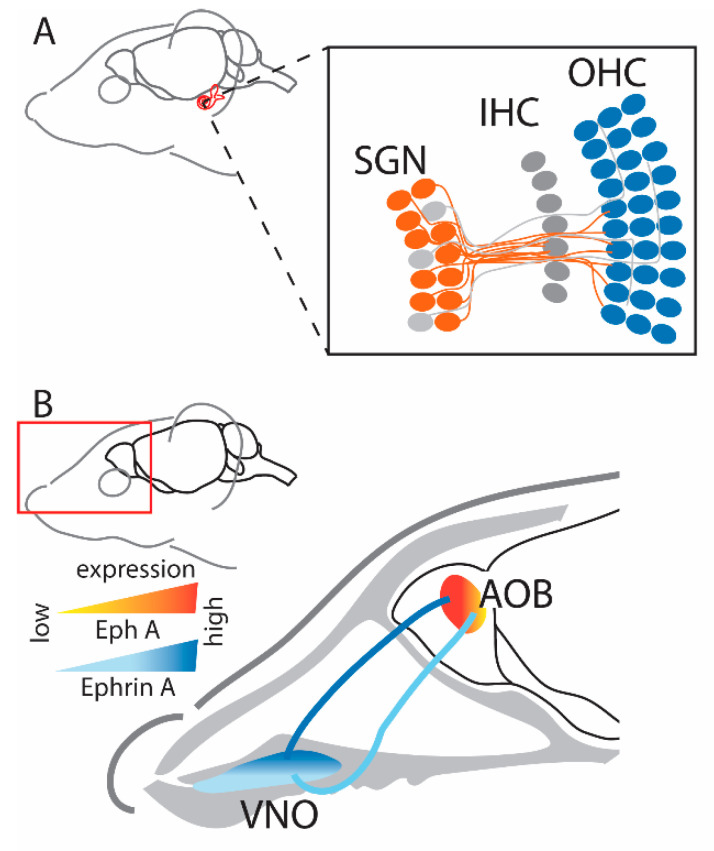
(**A**) Representation of the projections in the auditory system (cochlea) where type I SGNs expressing EphA4 project to OHCs that highly express ephrin-A5 ligands [55]. Note that their gradient is not well defined and therefore is not depicted here. Figure created using data from [55]. (**B**) Representation of the projections in the vomeronasal system where projections from VNO with EphA6 expression target regions with a similar expression of ephrin-A5 in the AOB [63], therefore unlike other sensory systems, projections in the vomeronasal system are guided by attraction. Figure created using data from [63] (SGN—spiral ganglion neurons; IHC—inner hair cells; OHC—outer hair cells; VNO—vomeronasal organ; AOB—accessory olfactory bulb).

### 2.4. Olfactory System

To this point, we have described mapping in sensory systems based on the repulsive interactions between Eph receptors and their ephrin ligands. However, a major exception to this pattern is the vomeronasal system, in which axons with high ephrin-A ligand expression project onto regions in the accessory olfactory bulb (AOB) with high EphA receptor expression [63]. In normal development, ephrin-A5 is expressed as a high-to-low gradient from apical to basal vomeronasal organ (VNO), while EphA6 has higher expression in the anterior AOB compared to the posterior part (Figure 3B); uniform expression of ephrin-A3 and Eph-A3 is also detected throughout these structures [63]. When investigating further using an in vitro stripe assay, it was found that in ephrin-A5^−/−^ mice, axons from the anterior VNO project to both anterior and posterior AOB [63]. These results suggest that, like other sensory systems, Eph/ephrin signaling is involved in establishing a map in the vomeronasal system. However, this might not be dependent on the repellent activity that is characteristic of Eph and ephrin interaction and instead be due to adhesion/attraction [64]. However, the exact nature of this interaction and how ephrins are involved in establishing the unique topographic organization of the vomeronasal system needs further investigation.

### 2.5. Integration of Disordered Projections in the Superior Colliculus

In addition to studying the organization of individual projections, there have been experiments examining the integration of multiple projections in the absence of EphA and ephrin-A, particularly within the SC. This region receives inputs from multiple sensory modalities and integrates sensory and motor information to guide appropriate behavioral responses.

One of the best-characterized examples of projection integration in ephrin-A mice is in the ipsilateral and contralateral visual input. Ipsilateral projections enter the SC later in embryonic development (E18) than contralateral ones, where they form a reversed map across the nasal-temporal retinal axis that integrates functionally with the contralateral retinal projections to facilitate binocular vision [65]. Anatomical tracing of the retinocollicular pathway has demonstrated that ephrin-A2^−/−^ and ephrin-A2/A5^−/−^ mice have a greater number of ipsilaterally projecting RGCs compared to wild-type mice [44] and like for the contralateral input, inappropriately located and ectopic diffuse ipsilateral terminals are present in the SC of ephrin-A2^−/−^ mice and more abundantly in ephrin-A2/A5^−/−^ mice [44]. This was confirmed by electrophysiological recordings from the superficial layer of the SC, where the receptive field size was larger and irregularly shaped in both strains suggesting overlapping and inaccurate TZs [44]. Furthermore, the contralateral and ipsilateral retinal inputs were misaligned, suggesting a failure of integration [44].

Similar results were found for the retinocollicular and corticocollicular maps within the SC. In wild-type mice, the retinocollicular map develops first in the upper stratum griseum superficiale (SGS) of the SC, followed by the corticocollicular map, which forms slightly deeper within the SC in the lower SGS, with both maps aligning to form retinotopic organization [41,65]. However, in ephrin-A^−/−^ mice, anatomical tracing demonstrated that there were more topographic errors in the retinocollicular map compared to the corticocollicular map in both ephrin-A2^−/−^ mice and ephrin-A2/A5^−/−^ mice [32]. Although a direct comparison of the maps was not carried out in the same animals, the discrepancy in the organization suggests that the two maps are not aligned within the SC [25]. This result matches studies in anophthalmic mice showing that the corticocollicular map can form even in the absence of the retinocollicular one and suggests that both retinal and cortical projections respond to SC guidance cues but may not be dependent on each other [66]. Furthermore, the lack of any visual activity in anophthalmic mice highlights that SC guidance cues are sufficient to establish the corticocollicular map in this model. However, the EphA3-KI mouse provides a different insight into the integration of maps within the SC. Although these mice have two functional maps in the SC, they only have one in the dLGN and V1 [41], raising the question of how the duplicate retinocollicular map in the SC integrates with the single map projecting from V1. Unexpectedly, it was found that a single injection of a fluorescent dye into the cortex resulted in two TZs in the SC, suggesting that the corticocollicular map aligned with the retinocollicular map, even though the latter was aberrant relative to the visual field input and the expression of ephrin-A in the SC. This model was also used to demonstrate that retinal activity is crucial for this map alignment, as deleting β2 cholinergic receptor activity in EphA3-KI mice resulted in only a single map in the corticocollicular projection that was misaligned with the duplicated retinocollicular projection.

While the ephrin-A2/A5^−/−^ and anophthalmic mouse models suggest that retinal and cortical visual inputs to SC are independent, studies in EphA3-KI mice suggest that they are not and that retinal activity is instrumental in aligning the maps. However, the situation is further complicated when considering the integration of different modality maps within the SC. Somatosensory (from the cortical barrel fields) and visual (retinal) maps project to the SC, with the somatosensory maps terminating deep within the SC, ventral to the visual input to the SGS. In ephrin-A2/A3/A5 triple-KO mice, the somatosensory map is abnormal, with larger and more diffuse TZs, and more numerous ectopic terminations compared to wild-type mice [67]. However, when integration of S1 and retinal visual projections to the SC was assessed in the EphA3-KI mouse, unexpectedly, the S1–SC projection did not align with the duplicated retinocollicular map, suggesting that unlike for the V1-SC projection, retinal input to the SC does not influence the topography of the S1–SC projection [67]. The result matches the normal sensory topography in monocular enucleation models [68,69,70] and also fits with the hypothesis that alignment requires activity-dependent mechanisms: the somatosensory and visual maps do not share activity patterns as they support different types of sensory information, and therefore are unlikely to use these cues for their integration. These data support a model in which different modalities establish topography within the SC using Eph and ephrin gradients, with activity-dependent mechanisms being involved in integrating converging inputs only when these inputs share a common source of activity (i.e., ipsilateral and contralateral eye, or cortical and retinal visual inputs, but not somatosensory and visual cortical inputs).

## 3. Linking Structure and Function: Abnormal Visual Behavior in Ephrin-KO Mice

Although the absence of guidance cues has an impact on multiple brain regions, the extensive characterization of the visual system, as well as the range of visual tasks that are available in mice, make vision a very attractive modality to relate structural changes to functional outcomes. In other words: how do these mice with altered visual maps see the world, and how do they respond to it?

Although there are defects along multiple pathways in the visual system of ephrin-KO mice, they have normal pupillary reflexes and visual acuity, with the only identified visual deficit being in visuomotor tracking behavior [44]. The selective visual deficit may be associated with the different severity of the topographic disorder in the retinocollicular and retinogeniculate pathways and/or the requirement for the integration of multiple sensory modalities in tracking responses.

The presence of normal visual acuity (~0.5 cycles per degree) in ephrin-KO mice is surprising because there is evidence that at least some of the abnormal and ectopic connections in the visual system of ephrin-A2^−/−^ and ephrin-A2/A5^−/−^ mice are functional and would therefore be expected to disrupt the continuity and/or coherence of visual input [34,44,71]. A possible explanation is that acuity is dependent on the retino-geniculo-cortical pathway [72], which is less disordered compared to the retinocollicular pathway in both these strains and may retain sufficient order to support normal acuity [32,33], already poor in wild-type mice [73,74]. Another consideration is that the behavioral task used to test acuity (forced-choice discrimination) [44] required animals to discriminate between a pattern (vertical stripes) vs. a solid grey panel or between horizontal and vertical stripes, tasks that could feasibly be achieved if the visual image were only partly fragmented by abnormal retinal projections [75]. It would be interesting to compare the behavioral acuity measurements with estimates of acuity using cortical recordings during the presentation of gratings to more precisely characterize the impact of local map disruptions within the cortical visual field.

The greater relative disruption in the retinocollicular and corticollicular pathways may be sufficient to explain the deficit in visuomotor tracking, a behavior that is mediated primarily by the SC [76]. In support of a role for visuotopic organization in tracking, the deficit was more severe in double-KO (ephrin-A2/A5^−/−^) compared to single-KO (ephrin-A2^−/−^) mice, in line with the extent of the topographic disorder [44]. Interestingly, the deficit was attributed to misaligned binocular projections: suturing one eye shut restored head tracking to the levels observed in wild-type mice [44]. These visuomotor deficits are therefore linked at least in part to the disruptions within the retinocollicular pathway but may also involve wider networks: the SC integrates binocular visual input with motor output, specifically with head orienting to a stimulus [72,77], and there is evidence that several of the sensory topographic maps within the SC are disrupted in ephrin-KO mice [41], potentially contributing to the abnormal motor output.

This example highlights a key limitation of many of the ephrin transgenic lines, which were produced before technology for conditional (temporal and spatial) control was widely available. Most are constitutive knockouts in which ephrin-A ligands are absent from the entire organism, and all lack ephrin-As from conception, which is likely to lead to compensatory reorganization, and/or indirect modifications to interconnected circuitry during development, making it difficult to definitively link behavioral deficits to abnormalities in a single circuit. Nonetheless, the intact visual acuity in ephrin-KO mice does provide an opportunity to use vision-based tasks to investigate the role of ephrins in brain regions involved in more complex behaviors. For example, learning strategies were assessed in a study with ephrin-A2^−/−^ mice using a visual discrimination task with reversal learning [78]. This study demonstrated maintenance of responding patterns in a reversal learning/set shifting task suggestive of perseverative behavior, implicating dysfunction in cortical and potentially striatal circuits [78]. Extending these results, ephrin-A2/A5^−/−^ mice were shown to have perseverative behavior on a progressive-ratio (PR) task [79], and ephrin-A2/A3^−/−^ mice showed repetitive self-injurious grooming behavior [80]. Moreover, ephrin-KO mice also show altered social behaviors, such as reduced inter-male aggression [81] and decreased maternal behavior [82]. The behaviors are complex and involve pathways including the orbitofrontal cortex, thalamus, striatum, and prefrontal cortex, and have been proposed to be relevant for investigating aspects of ASD and obsessive-compulsive disorder (OCD) [80].

Although several studies have focused on visual behavior in ephrin-KO mice, these mouse models may allow exploration of how disruptions in sensory pathways might influence the development of downstream circuits, a topic that is gaining increasing interest as research into human neuropsychiatric conditions like ASD and schizophrenia are linked to sensory abnormalities [4,5]. However, as described above, in constitutive knockout animals, it is not possible to differentiate between the direct impact of ephrin removal on the organization of cortical/reward pathways and indirect changes to cortical circuitry as a result of abnormal sensory input. It would be interesting to look at strains with spatially and/or chronologically controlled transgene expression. For example, EphA3-KI mice [46], in which abnormal expression is primarily limited to the retina, have perseverative behavior or self-injurious behavior, or is it the impact of abnormal visual projections limited to sensory responses. Development of conditional transgenic lines where an Eph receptor or ephrin ligand is removed specifically from one structure (e.g., the retina or the cortex) or at a particular developmental timepoint would provide further information on how topographic abnormalities in sensory pathways might impact on whole brain circuitry.

## 4. What Do Ephrin-KO Mice Tell Us about Activity-Dependent Refinement?

Although this review has focused primarily on the role of ephrin-A ligands in establishing topography, the formation of mature functional maps also requires neuronal activity [7]. Ephrin-A-KO mice can therefore serve as a model to explore the capacity for neuronal activity to establish a map in the absence of ephrin cues. This question has therapeutic relevance in designing interventions for neurological conditions involving abnormal connectivity.

Within the visual system of ephrin-KO mice, the presence of ectopic terminations is a key phenotype, and there is evidence from a range of techniques that these ectopic terminations are functional [34,43,44,83]. Interestingly, computational modeling based on optical imaging experiments suggests that individual ephrin-KO mice have unique patterns of ectopic projections, implying that the errors in the map are sculpted by individual visual experience and are not hardwired [43]. Perhaps surprisingly, these ectopic connections persist despite activity-dependent mechanisms that would normally remove them during the critical period [84]. The ephrin-A-KO mouse, therefore, presents a useful model in which to test potential therapeutic interventions that increase activity-dependent processes to drive plasticity and repair abnormal connectivity. While the β2 nAChR-KO model described earlier (Section 2.1.1) provides proof of principle that modifying activity-dependent processes in the absence of ephrins can impact topography, a non-invasive intervention to increase activity would provide more direct therapeutic evidence. There is a range of interventions that can alter visual system activity, including environmental enrichment, modification of the light environment, pharmacological interventions, and direct modulation of neuronal activity through neuromodulation. Among these, a series of experiments have investigated the use of repetitive transcranial magnetic stimulation (rTMS) to increase activity and demonstrated partial repair of abnormal connections [85].

## 5. TMS as a Method to Increase Activity and Compensate for Loss of Ephrins

rTMS is a non-invasive brain stimulation technique that has been used as both a therapeutic and experimental research tool. rTMS can modulate cortical activity by generating a pulse of a magnetic field which induces an electrical current in the targeted brain region [86]. The ability of rTMS to excite or inhibit cortical and subcortical activity [87,88,89] and its extended effects on distributed networks across the brain [90,91,92,93] has made it a promising therapeutic tool, and it has been approved for the treatment of disorders such as depression [94] and OCD [95]. There are other techniques, such as environmental enrichment [96,97,98,99] and transcranial direct current stimulation (tDCS) [100,101], that have been used to manipulate neural activity in rodent models. However, due to limited evidence, they have not been discussed in this review.

To gain a better understanding of how rTMS affects the brain, custom miniature coils have been developed that deliver focal low-intensity (LI) magnetic stimulation ranging between 1 and 150 mT to the rodent brain [85,102,103]. These coils were first used in ephrin-KO mice to attempt to increase neuronal activity and repair abnormal topography [104]. Most studies of LI-rTMS in ephrin-KO mice use biomimetic high-frequency stimulation (BHFS) patterns where pulses are delivered at frequencies between 6–10 Hz that mimic endogenous patterns of electrical fields around activated nerves during exercise [105].

These initial experiments applying LI-rTMS to ephrin-KO mice showed a reduction in the number of ectopic retinocollicular terminals in adult ephrin-A2/A5^−/−^ mice following chronic LI-rTMS and an increase in receptive field size in the SC [104]. In subsequent studies, LI-rTMS showed network-wide effects, including an improvement in corticocollicular topography and lower dispersion of retrogradely-labeled dLGN neurons in adult ephrin-A2/A5^−/−^ mice [106,107]. The improvement in visual circuitry following LI-rTMS is also accompanied by improved visual tracking behavior [104,107]. In these studies, LI-rTMS did not disrupt the normal visual circuitry, as no significant changes were observed in wild-type mice following treatment. Therefore LI-rTMS might selectively target the abnormal ectopic terminals to improve the structure and function of the visual system in ephrin-KO mice.

One of the mechanisms behind this was suggested to be the susceptibility of small and weak connections to activity-dependent mechanisms: they express NMDA receptors, may be silent synapses, and are more likely to undergo long-term depression (LTD) and be removed or altered when they are activated [104,106]. Interestingly, a single stimulation of LI-rTMS caused an increase in the number of ectopic projections that were detected functionally, suggesting that the activity induced by rTMS could be activating silent synapses and making them susceptible to removal through visually driven activity-dependent refinement [104]. Consistent with this hypothesis, some effects of LI-rTMS appear to require concurrent sensory-evoked cortical activity. LI-rTMS effects on cortical visual evoked potential (VEP) characteristics and corticocollicular circuitry in ephrin-A2/A5^−/−^ mice were abolished when stimulation was delivered in the dark [83,108]. However, it is likely that some of these effects are pathway-specific because the reorganization of geniculocortical projections by LI-rTMS was not prevented in a dark environment [108]. This suggests that in some circuits, normal vision or rTMS alone is not sufficient to reorganize abnormal topography, but together, they partially rescue the phenotype. It isn’t clear whether this is an additive effect: more activity results in better refinement or if the two interventions act via distinct mechanisms. However, additional improvement was observed when rTMS was delivered in the light and concurrently with physical exercise [108], suggesting that it is possible to manipulate brain activity to increase the strength and impact of activity-dependent plasticity processes, even in the adult brain.

The removal of silent synapses could also be dependent on other mechanisms, such as the changes in levels of neurotrophic factors following rTMS. Brain-derived neurotrophic factor (BDNF) acts via the p75 neurotrophin receptor (p75^NTR^) in activity-dependent axonal pruning via programmed cell death processes [109], which explains the removal of ectopic terminals along the retinocollicular pathway [104] where p75^NTR^ is highly expressed [110,111]. Another factor contributing to this removal of ectopic terminals could be the elevated expression of neuronal nitric oxide synthase (nNOS) following LI-rTMS [104]. nNOS is involved in refining and developing retinocollicular projections as NO induces LTD in aberrant connections [112]. However, removal of ectopic terminals in ephrin-KO mice following LI-rTMS was only seen along the retinocollicular pathway and not the corticocollicular and geniculocortical pathways where the circuitry was reorganized instead [104,106]. This could be due to the activation of tropomyosin receptor kinase b (TrkB) by BDNF along corticocollicular and geniculocortical pathways, as BDNF having a higher affinity for TrkB inhibits p75^NTR^-mediated apoptosis along these pathways and instead promotes reorganization by refining axonal arbors [113,114]. rTMS-induced changes in BDNF to improve neural circuitry in the adult brain is similar to the upregulation of BDNF during the “critical period” of development to promote neural plasticity [115], which could suggest that rTMS might be unlocking juvenile-like plasticity in the adult brain. It will be important in future experiments to use other interventions such as environmental enrichment [116] or plasticity-promoting drugs such as fluoxetine [117] to further explore the limits of neuronal activity in repairing abnormal connections.

## 6. Conclusions

Transgenic animal models with manipulations of Eph and ephrin expression have provided us with unexpected insights into the mechanisms behind brain organization, highlighting the roles of molecular guidance cues and their interactions with neural activity in the development and plasticity of topographic maps in the brain. These models have also allowed an exploration of the link between brain organization and behavior in simple and complex tasks, uncovering exciting parallels between ephrin-KO models and models of neuropsychiatric disorders. Evidence in these mice showing that abnormal connectivity can be repaired using neuromodulation aligns with evidence from human TMS neuroimaging studies showing the distributed impact on multiple brain circuits [85,90,93]. These models will be a useful part of future investigations into task-based and pharmacological interventions that could be combined with TMS to improve brain connectivity and behavioral changes.

## Figures and Tables

**Figure 1 ijms-24-06966-f001:**
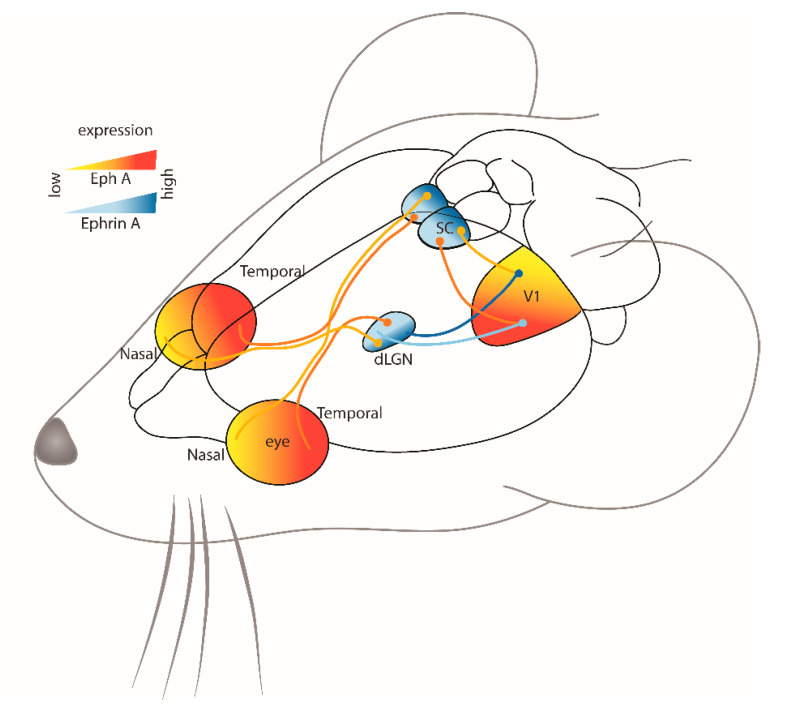
Schematic representation of the visual system circuitry guided by Eph-ephrin gradients in different regions of the adult mouse brain. The endpoints of the projections are represented by a circle (dLGN-dorsolateral geniculate nucleus; SC-superior colliculus; V1-primary visual system).

**Figure 2 ijms-24-06966-f002:**
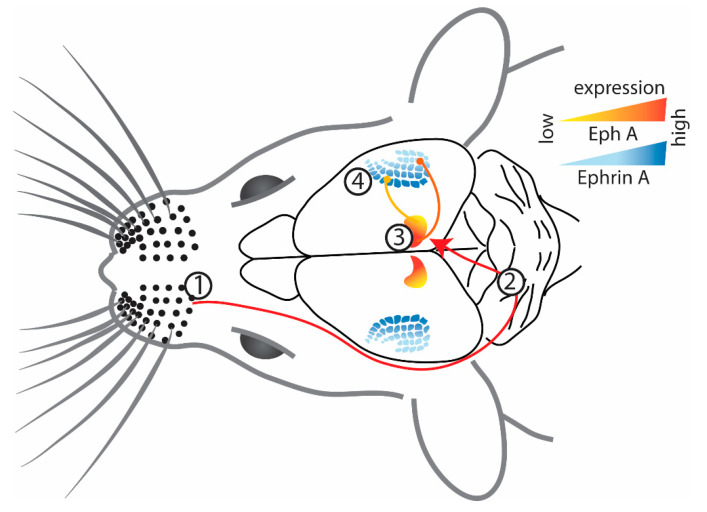
Schematic representation of the projections from (1) whisker barrels to (2) brainstem, (3) thalamus, and to (4) contralateral barrel cortex in adult mice [47]. Ventrobasal thalamus projects to the barrel cortex, which is guided by their complementary gradient of EphA4 receptor and ephrin-A5 ligands, respectively [49]. The endpoints of the projections are represented by a circle. Figure created using data from [47,49].

## Data Availability

Not applicable.

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
