# Peer review of "Interactions between Guidance Cues and Neuronal Activity: Therapeutic Insights from Mouse Models"

_ijms, 2023, doi:10.3390/ijms24086966_

Round 1
Reviewer 1 Report
I would suggest to check that the font is consistent through the manuscript. Sometimes in some part the font is smaller.
Author Response
We would like to apologise for the inconsistency in the format of our manuscript. We think the font size changed when the submission portal converted our original manuscript to the journal template. We have changed the format of the manuscript and made the font consistent.
Reviewer 2 Report
In this review manuscript, the authors have summarized and discussed the brain maps with a focus on the Eph and ephrin families of axon guidance cues using transgenic models. There are several points the authors need to address to improve the manuscript.
1. As stated by the authors, the selected transgenic models have allowed an exploration of the link between brain organization and behavior. However, only abnormal visual behaviours had been discussed. Are any other behavioural changes have been studied?
2. The author had discussed only one method, TMS, to increase activity and compensate for loss of ephrins. To improve the quality of this review manuscript, more strategies, like pharmacological interaction, to improve brain connectivity and behavioral changes are encouraged to included.
3. The writing of this manuscript should be carefully improved. Lots of typos and fonts problems could be found in this manuscript, e.g. Page 11.
Author Response
Point 1: As stated by the authors, the selected transgenic models have allowed an exploration of the link between brain organization and behavior. However, only abnormal visual behaviours had been discussed. Are any other behavioural changes have been studied?
Response 1: Thank you for your comment. We have discussed more about visual behaviour as that has been linked with structural abnormalities in ephrin-KO mice in several studies. We have explored other behavioural changes in ephrin-KO mice and discuss them briefly in section 3 from line 527-535.
“Extending these results, ephrin-A2/A5-/- mice were shown to have perseverative behavior on a progressive-ratio (PR) task (79) and ephrin-A2/A3-/- mice show repetitive self-injurious grooming behavior (80). Moreover, ephrin-KO mice also show altered social behaviors such as reduced inter-male aggression (81) and decreased maternal behavior (82). The behaviors are complex and involve the pathways including the orbitofrontal cortex, thalamus, striatum and prefrontal cortex, and have been proposed to be relevant for investigating aspects of ASD and obsessive-compulsive disorder (OCD) (80).”
Point 2: The author had discussed only one method, TMS, to increase activity and compensate for loss of ephrins. To improve the quality of this review manuscript, more strategies, like pharmacological interaction, to improve brain connectivity and behavioral changes are encouraged to included.
Response 2: We have focussed on TMS in our review as there is more evidence of its use in ephrin-KO mouse models. However, we acknowledge there are other treatments such as pharmacological interventions and neuromodulation that have therapeutic relevance. Therefore, we have briefly discussed these interventions in section 5 of our review.
Point 3: The writing of this manuscript should be carefully improved. Lots of typos and fonts problems could be found in this manuscript, e.g. Page 11.
Response 3: We would like to apologise for the inconsistency in the format of our manuscript. We think the font size changed when the submission portal converted our original manuscript to the journal template. We have changed the format of the manuscript and corrected the typos and font style.
Reviewer 3 Report
I have to say that this review, although not systematic, is very well written and interesting. From my point of view it can be acceptable as it is. The only comment I would make is that there are indications that Ephrin-B1/2 may play a role not only in development but also in adult functional plasticity linked to learning and functional recovery. This could be stated somewhere in the introduction (for example PMID: 29444449; 22101302; 31633063; 32256333; 26928051).
Author Response
Thank you for your comment. Although we have not focussed on ephrin-B in our review, we agree its role in neural plasticity is worth mentioning so we have included it in our introduction from line 72-76.
“EphB and ephrinB families also have important roles in visual system development (particularly axonal decussation (12) and dorso-ventral patterning (13)), and in the hippocampus (14–18). However, in this review we focus on the ephrin-A family because of their key role in topographic mapping and integration of topographic projections.”